# The Effects of Nitrate Supplementation on Performance as a Function of Habitual Dietary Intake of Nitrates: A Randomized Controlled Trial of Elite Football Players

**DOI:** 10.3390/nu15173721

**Published:** 2023-08-25

**Authors:** Matjaž Macuh, Nenad Kojić, Bojan Knap

**Affiliations:** 1Department of Food Science and Technology, Biotechnical Faculty, University of Ljubljana, Jamnikarjeva 10, 1000 Ljubljana, Slovenia; 2Department of Nephrology, University Medical Centre Ljubljana, Zaloška 7, 1000 Ljubljana, Slovenia; 3Faculty of Medicine, University of Ljubljana, Korytkova ulica 2, 1000 Ljubljana, Slovenia

**Keywords:** nitrates, dietary supplements, sport performance, nitric oxide, physical activity, beetroot, football

## Abstract

Nitrates are an effective ergogenic supplement; however, the effects of nitrate supplements based on habitual dietary nitrate intake through diet alone are not well understood. We aimed to assess this in a group of 15 highly trained football players from Slovenian football’s First Division. Participants underwent two separate Cooper performance tests either with nitrate supplementation (400 mg nitrates) or placebo while having their nutrition assessed for nitrate intake, as well as energy and macronutrient intake. Nitrate supplementation had a statistically significant positive effect on performance if baseline dietary nitrate intake was below 300 mg (*p* = 0.0104) in both the placebo and intervention groups. No effects of nitrate supplementation when baseline dietary nitrate intake was higher than 300 mg in the placebo group could be concluded due to the small sample size. Nitrate supplementation did not have a significant effect on perceived exertion. The daily nitrate intake of the participants was measured at 165 mg, with the majority of nitrates coming from nitrate-rich vegetables.

## 1. Introduction

Nitrates, found predominantly in green leafy and root vegetables, have been studied extensively for their possible ergogenic benefits through a wide variety of sports modalities and exercise lengths [1,2,3,4]. Through the formation of nitric oxide (NO) via the nitrate–nitrite–nitric oxide metabolic pathway, nitrates augment endurance performance by decreasing oxygen cost during submaximal aerobic exercise and increase power output through enhanced muscle contractile function [5]. The latter has positive effects on improved exercise tolerance, which might be influenced by the effects of nitrates on decreased perceived effort to exercise observed in certain studies [6]. These effects are likely most prominent when nitrates are taken 2–3 h prior to exercise of 10–16 min duration, although positive effects of nitrate supplementation have been documented in wider exercise time frames [1,3]. Nitrates exhibit ergogenic benefits when taken either acutely or chronically with the effective dosage being 5–16.8 mmol, corresponding to 300–1041 mg of nitrates [2,3,4].

The formation of nitric oxide is stimulated either through the ingestion of nitrate-rich foods, and to a certain degree through endogenous synthesis through the oxygenation of the amino acid L-arginine to citrulline, followed by the oxygenation of citrulline to nitric oxide [7,8]. The endogenous pathway of nitric oxide formation is relatively limited as approximately one bowl of green leafy vegetables supplies larger quantities of nitrates than are produced endogenously throughout an entire day [9]. Athletes seeking to increase sports performance via nitrates should thus be encouraged to cover their nitrate needs through a nitrate-rich diet in the form of nitrate-rich vegetables and/or supplements. Concentrated beetroot shots or beetroot juice are one of the most widespread food sources used, with beets having relatively high nitrate content [10]. Although dietary supplements might represent the more convenient route of achieving sufficient nitrate intake to elicit ergogenic benefits, using supplements has its drawbacks. Firstly, supplements can be problematic due to contamination with prohibited substances, which is not only alarming from a health perspective, but might also be the cause of an athlete failing a doping test [11,12]. Nutritional supplements also represent an additional financial burden for athletes who often operate on a limited budget. Furthermore, covering nitrate needs through whole foods might offer additional benefits beyond nitrate intake alone by enriching an athlete’s diet with certain vitamins, minerals, and fiber, all of which can be found in nitrate-rich vegetables [13]. Consequently, meeting one’s nitrate needs from whole foods rather than supplementation should most likely be the goal of every athlete, or at least through the majority of an athlete’s training period.

While many studies have examined the effectiveness of nitrates on different aspects of physical performance [1,2,3,4], the research area of covering sufficient nitrate intake to elicit ergogenic benefits from whole foods alone is not well studied. Jonvik et al. (2017) reported a median dietary nitrate intake of 106 mg nitrates daily in a group (n = 553) of highly trained Dutch athletes [14], which is below the established ergogenic dose of at least 300 mg [2,3,4]. Several authors have made practical recommendations on achieving ergogenic doses of nitrates through whole foods. For example, Van der Avoort et al. (2018) reported that an intake of approximately 350 mg of nitrates could be achieved by eating 175 g of lettuce, 200 g of spinach, 115 g of pak choi, 80 g of rocket salad/arugula, or combinations of these products [13]. Jonvik et al. (2017) provided similar recommendations of achieving an intake of 400 mg of nitrates in the form of 200 g of lettuce, 222 g of spinach, 123 g of pak choi, or 95 g of rocket salad/arugula [14]. Hord et al. (2009) reported an estimated nitrate intake of at least 175 mg and up to 1200 mg when 4–5 servings of both fruit and vegetable are included in the diet [15]. However, all of these recommendations are educated guesses made based on the average nitrate content of nitrate-rich vegetables. Clearly, daily nitrate ingestion through nitrate-rich foods varies depending on several factors, such as the type of vegetable consumed, the type of harvesting and cultivation, as well as the food preparation type [16,17]. Altogether, these numbers indicate that eating enough nitrates through whole foods to achieve ergogenic benefits is feasible; however, this might not always be attainable. For instance, only 2% of athletes in the Jonvik et al. (2017) trial had a habitual nitrate intake exceeding 400 mg, which is not only an effective ergogenic dose, but often a standard supplement dose as well [14]. There is also no evidence of whether the effect of supplementing the diet with nitrates on performance in athletes is less effective if the athlete already consumes a greater proportion of foods richer in nitrates despite not meeting the minimal effective dose of nitrate intake of 300 mg reported in other studies [2,3,4]. The latter can be observed in the case of certain supplements. For example, in the case of creatine, populations with the lowest intake of creatine from the diet (i.e., vegetarians and vegans) experience the most pronounced effect of creatine supplementation on total creatine levels, lean mass, and total work volume [18]. How this effect is reflected in the case of nitrates and where the limit of dietary nitrate intake is, where the addition of a dietary nitrate supplement offers no further significant benefits, are still unexplored.

Previous research indicates that nitrate supplementation is less effective in well-trained individuals, defined in this case as athletes having a maximal oxygen uptake (VO2max) below 65 mL/kg/min [7,19]. This could be explained by the fact that elite athletes have higher baseline nitrate levels when compared to less trained individuals [4,20], making them less dependent on nitric oxide production via the nitrate–nitrite–nitric oxide pathway. Another explanation is that these athletes might have a higher habitual nitrate intake from nitrate-rich foods, corresponding to healthier eating patterns often seen in these populations [14], and/or a higher energy intake and, thus, a higher nitrate intake [21].

Thus, the purpose of this article was to find out the effects of nitrate supplementation on performance depending on different baseline levels of athletes’ habitual dietary nitrate intake on a sample of elite football players from the First Division of Slovenian football. Based on this, the following hypotheses were determined:

**Hypothesis** **1.**
*Nitrate supplementation will have a significant positive effect on covered distance in the Cooper performance test compared to placebo if the baseline dietary nitrate intake is low (<300 mg nitrates) in both the placebo and intervention groups.*


**Hypothesis** **2.**
*Nitrate supplementation will not have a significant positive effect on covered distance in the Cooper performance test compared to the placebo if the baseline dietary nitrate intake is high (>300 mg nitrates) in the placebo group.*


**Hypothesis** **3.**
*Perceived exertion during each quarter of the trial, as well as average perceived exertion, will be significantly lower in the intervention group compared to placebo if the baseline dietary nitrate intake is low (<300 mg nitrates) in the placebo group.*


An amount of 300 mg of baseline dietary nitrate intake was chosen as a reference value for performance comparison, as previous research has reported that at least 300 mg of nitrates is needed to improve exercise performance [2,3,4]. Additionally, the dietary intake of nitrates, where there were no statistically significant differences in the covered distance in the Cooper performance test between the placebo and intervention groups, was examined to give further insight as to how nitrate supplementation affects performance based on baseline nitrate intake.

The main objective of this study was to examine the effects of nitrate supplementation on performance as a function of the habitual dietary intake of nitrates on a sample of elite Slovenian football players, with perceived exertion being the secondary response variable. The main conclusions of this study are that nitrate supplementation augments performance when dietary nitrate intake is below 300 mg, with nitrates having no significant effect on perceived exertion during exercise. The average nitrate intake of Slovenian First Division football players was 165 mg, with nitrates being consumed mostly from leafy vegetables.

## 2. Materials and Methods

This was a randomized, placebo-controlled, double-blind, crossover trial consisting of two phases: introductory testing and the first performance assessment (Cooper test) and the second performance assessment. Subjects were randomly assigned into one of two groups (A and B) using simple randomization with Excel by an outside party within the research institution; the details of the randomization series were unknown to any of the investigators or the coordinator. The details of the randomization were kept in a secure place (personal laptop secured with a personal code) by the researcher. Based on the randomization, the researcher labeled the provided supplements and placebo drinks accordingly to be used in the intervention trial. After all relevant data were gathered and the trials were finished, we gained access to data as to which football players received nitrate supplementation in the first trial and placebo in the second trial prior to the performance assessment, and which football players received placebo supplementation in the first trial and nitrate supplementation in the second trial prior to the performance assessment. Details around the performance assessment and supplementation strategies are explained in Section 2.2 (Performance Assessment). A schematic overview of the research process is presented in Figure 1. Ethical approval was given by the Medical Ethics Commission of the Republic of Slovenia (protocol code 0120-601/2021/15).

### 2.1. Subjects

Sample size was calculated using online software developed by Schoenfeld [22], using the formula designed for cross-over studies. The significance level was set at 0.5, and power at 0.9. The within-patient standard deviation was set at 100 based on our assumption. The minimal detectable difference in means was set at 90 based on previous research showing an approximately 3% improvement in similar performance tests in athletes with nitrate supplementation [1,3], and based on the assumption that football players would cover on average 3000 m in the Cooper performance test. Based on these numbers, a total of a total of 10 participants was determined necessary for this study. Assuming a dropout rate of 20% and the fact that daily nitrate intake was unknown but was an important factor for further investigation, we determined that 15 participants should be included in this study.

Thus, 15 football players aged between 19 and 31 were included in this study. Subjects participated in the study voluntarily and could withdraw from the study at any point. All football players were members of the same club that played in the First Division of Slovenian football. This sample was chosen as we expected it to be quite homogeneous, as the footballers had similar height and weight, level of physical fitness, and training regimen, and they were consuming two similar meals daily at the football club. The exclusion criteria were the following: all football players with a history of cardiovascular disease and high blood pressure and/or using medications that could affect the outcome of the study, injured athletes, and athletes with a history of nitrate usage less than 6 months prior to the start of the study. The testing protocol was explained to the participants and they familiarized themselves with the testing equipment prior to the first Cooper test. Two hours before the start of the first Cooper trial, the participants consumed 70 mL of concentrated beetroot juice (nitrate content: 400 mg; manufacturer: Beet It, Nitrate 400 Sport Shot, James White Drinks, Ipswich, UK) in a single dose. Alternatively, the participants consumed an equivalent amount of placebo that looked and tasted similar to the aforementioned supplement, but with nitrates removed (nitrate content ≈ 0 mg; manufacturer: Beet It, Placebo Shots, James White Drinks, Ipswich, UK). The participants then underwent bioimpedance body composition measurements (BIA 101 BIVA^®^ PRO, Class IIa Medical Device—93/42/EEC Class IIa Medical Device—93/42/EEC, Akern, Pisa, Italy) and completed a 3-day food diary. Two weeks after the first test, the participants returned to allow for a sufficient wash-out period for nitrates [23]. During this time, the participants were asked to keep their diet the same and not use a mouth rinse which could blunt oral nitrate reduction [24]. The participants then returned for their second testing, where they consumed either the nitrate or placebo supplement (depending on the first trial), completed the food diary again, and repeated the Cooper test.

### 2.2. Performance Assessment

Sports performance was assessed using the Cooper test [25]. The maximum distance participants could cover in 12 min was measured. This test was chosen because the duration of the test corresponds to the time when nitrate supplementation has the greatest positive effect on physical performance [3]. The results of the Cooper test were also used to determine the approximate VO2max results of each participant via the equation [25]:VO2max (mL/kg min) = (22.351 × distance covered in meters) − 11,288

A treadmill (LifeFitness 95Ti model, Life Fitness, Ljubljana, Slovenia) was used for the Cooper test to ensure that the conditions were as similar as possible for all subjects in terms of weather conditions. Prior to testing, the participants were given 60 min to prepare for the test and to warm up. The warm up consisted of 10–15 min of easy running. During this period, the participants self-selected the speed at which they later began testing. This was followed by a period of 10–15 min when the participants individually prepared for the test, including foam rolling, stretching, and other warm up strategies they were accustomed to performing for training and/or matches. Two hours after consuming nitrates or placebo, the participants began the Cooper test. This allowed enough time for the conversion of nitrates to nitric oxide [3]. The participants were able to change the speed of the treadmill during testing. No foods or drinks were allowed during the testing period, and caffeine was prohibited prior to testing. During running, the participants gave their subjective perception of fatigue during exercise measured using Borg’s 15-point scale (Appendix B) at each quarter of the trial (after three, six, nine, and twelve minutes of running) [26]. After completing the test, the participants could continue running at a lower intensity for a while to cool down. A qualified professional as well as a doctor was also present throughout the testing, so that the process was carried out safely under expert guidance and in a safe environment.

### 2.3. Nutritional Assessment

The dietary intake of football players was assessed using 3-day food diaries (Appendix C). Diaries were provided with detailed instructions and an example of how to log food to avoid incorrect or inaccurate logging. The subjects were asked to provide descriptions that were as detailed as possible of the foods and liquids consumed as well as any nutritional supplements used. The subjects were asked to pay special attention to assembled dishes at home or in restaurants (e.g., risottos, meat platters, or soups) and to photograph them if possible. Additionally, the subjects were given the researchers’ contact information and an initiative to contact the research team if they had any questions regarding the process and food logging. Three-day food diaries were used for nutritional assessment where participants reported their eating habits on three consecutive days prior to both Cooper tests. After completing the food diaries, each subject sat down with one of the researchers to revise the completed diaries.

Food diaries were used to assess average nitrate intake prior to both Cooper tests and to evaluate daily energy and macronutrient intake prior to both trials, as a high variation in energy and nutrient intake could influence subsequent performance.

### 2.4. Body Composition Assessment

The body mass and height of the subjects were measured using a scale with a stadiometer. Based on these data, the body mass index (BMI) was calculated. The participants underwent a bioelectrical impedance vector analysis (BIA 101 BIVA^®^ PRO, Akern, Pisa, Italy) to obtain body composition parameters. Fat mass (FM), fat-free mass (FFM), and total body water (TBW) were obtained using Bodygram PLUS Software V. 1.0 (Akern Srl., Pontassieve, Florence, Italy). Furthermore, as BIVA represents a qualitative analysis [27], the raw data, such as the resistance and reactance of each participant, are available as Appendix A.

### 2.5. Statistical Analysis

All data obtained from food diaries, Cooper tests, and body composition measurements were statistically analyzed using Excel 2016 (Microsoft Office, Redmond, WA, USA) and the statistical analysis programming language R (R version 4.1.0). Descriptive statistics were determined: minimum (min) and maximum (max) value, mean value (*x*), and standard deviation (SD). A paired-samples t-test was determined suitable to test for differences in the average covered distance on the Cooper performance test between both treatments, as well as differences in the average perceived exertion during each quarter of the trial. Our samples were paired since data were collected from assessing performance twice on the same participant. Additionally, we confirmed that the results of both groups and differences between groups followed a normal distribution for all the covered distances in the Cooper performance test as well as the average perceived exertion during each quarter of the trial.

## 3. Results

### 3.1. Sample Characteristics

Recruitment of the participants commenced at the beginning of the summer break of the First Division of Slovenian football in consultation with the club management. The sample size included 15 professional football players from the First Division of Slovenian Football. The participants’ characteristics are presented in Table 1. All 15 participants underwent randomization, performance testing with nitrate supplements and placebo, as well as other measurements, and thus successfully completed the experimental trial. There were no exclusions. The experimental trials were conducted during the same summer break to ensure the participants were not in the competition period and could thus perform at their best. The experimental trial followed the schematic as highlighted in Figure 1 and took two weeks to complete which includes the wash-out period.

### 3.2. Dietary Intake Evaluation

The dietary habits of football players were assessed using 3-day food diaries. The data were then analyzed using the PRODI^®^ 6.4 Expert program and R statistical processing programming language (R version 4.1.0). The nutritional intake of the subjects is presented in Table 2.

There were no statistically significant differences between and within participants for the placebo and intervention in all the variables measured (*p* > 0.05).

### 3.3. Results of Hypothesis Testing

#### 3.3.1. Hypothesis 1

Hypothesis 1 proposed that nitrate supplementation would have a significant positive effect on covered distance in the Cooper performance test compared to placebo if the baseline dietary nitrate intake was low (<300 mg nitrates) in both the placebo and intervention groups. In 13 participants, the baseline nitrate intake was below 300 mg in both the placebo and the intervention groups, with two players exceeding these baseline values and being omitted from Hypothesis 1. The football players in the intervention group had an average covered distance in the Cooper performance test of 2.98 km, with a standard deviation of 0.28 km, minimum of 2.48 km, and a maximum of 3.52 km. The average distance covered by the placebo group was 2.84 km, with a standard deviation of 0.23 km, minimum of 2.4 km, and a maximum of 3.5 km (Figure 2). Hypothesis 1 was accepted using a paired-samples t-test. We found a statistically significant difference in covered distance in the Cooper test between the placebo and intervention groups (t = −3.0336, df = 12, *p*-value = 0.0104). A sample estimate of the mean differences between pairs is −0.1446154 km with a corresponding 95% confidence interval [−0.25 −0.04].

#### 3.3.2. Hypothesis 2

Hypothesis 2 proposed that nitrate supplementation would not have a significant positive effect on covered distance in the Cooper performance test compared to the placebo if the baseline dietary nitrate intake was high (>300 mg nitrates) in the placebo group. Only two participants were eligible for this hypothesis testing, labeled as participant numbers 14 and 15 (Figure 3). The football players in the intervention group had an average covered distance in the Cooper performance test of 3.01 km, with a minimum of 2.94 km and a maximum of 3.08 km. The average distance covered by the placebo group was 3.02 km, with a minimum of 2.95 km and a maximum of 3.09 km (Figure 3). Based on the low number of eligible participants, the statistical test for this hypothesis was omitted as no definitive conclusions could be drawn.

#### 3.3.3. Hypothesis 3

Hypothesis 3 proposed that perceived exertion during each quarter of the trial, as well as average perceived exertion, would be significantly lower in the intervention group compared to placebo if the baseline dietary nitrate intake was low (<300 mg nitrates) in the placebo group. Hypothesis 3 was rejected using a paired-samples t-test. There was no statistically significant difference in average perceived exertion in the Cooper test between the placebo and intervention groups in the first quarter (t = −0.39477, df = 12, *p*-value = 0.6999), second quarter (t = −0.74278, df = 12, *p*-value = 0.4719), third quarter (t = 0.2907, df = 12, *p*-value = 0.7762), or fourth quarter of the Cooper test (t = 0.56195, df = 12, *p*-value = 0.5845) (Figure 4). As perceived exertion did not statistically differ at any time point between the placebo and intervention trial, analyzing the difference between mean average perceived exhaustion throughout the trial was deemed unnecessary.

## 4. Discussion

This study aimed to illuminate the effects of nitrate supplementation on performance depending on different baseline levels of athletes’ habitual dietary nitrate intake, as well as how athletes could cover their nitrate needs from whole foods alone without the use of nutritional supplements, on a sample of elite football players from the First Division of Slovenian football.

The results of this study indicate that nitrate supplementation has a statistically significant positive effect on performance if dietary nitrate intake is below 300 mg, which is regarded as an effective ergogenic dose [2,3,4]. When comparing performance on the Cooper test in participants who had a nitrate intake below 300 mg, the addition of a nitrate supplement significantly improved the covered distance in the Cooper test performance by an average of 0.145 km with a corresponding 95% confidence interval [0.0407–0.248] (*p*-value = 0.0104). The average nitrate intake for these participants in the intervention group was 142.2 mg and 142.8 mg in the placebo group, with all participants but one improving their performance in the Cooper test following nitrate supplementation. This would indicate that approximately 140 mg of nitrates ingested through whole foods is not enough to elicit the full ergogenic potential of nitrates and that athletes having a nitrate intake equal to or below this amount would likely benefit from additional nitrate supplementation.

Overall, the results of this study are in line with previous reviews [1,2,3,4,5] and the position statement by the IOC consensus statement [28] regarding the beneficial benefits of nitrates on exercise performance. The intervention group had, on average, improved running distance in the Cooper performance by 0.145 km with a corresponding 95% confidence interval [0.0407–0.248] (*p*-value = 0.0104). This was a 5% improvement in exercise performance with nitrate supplementation, which is in line with previous research on this topic reporting improvements of 4–25% in time-to-exhaustion performances, and improvements of 1–3% in sport-specific time trials lasting less than 40 min [29,30]. Comparing these results specifically to performance among football players, Nyakayiru et al. (2017) reported improvements in covered distance in the Yo-Yo trial following ~800 mg of nitrate supplementation for 6 consecutive days [31]. The improvements reported by the authors (3.4 ± 1.3%) were similar to those observed in this research, although there were distinct differences between the performance tests used (Yo-Yo versus 12-min Cooper test).

Looking at individual numbers, there was some conflicting evidence to be observed. For instance, participant 3 (Figure 2) had an average dietary nitrate intake of 248 mg for the placebo treatment and 268 mg for the intervention treatment, which was the highest nitrate intake within the intervention trial that was still below the ergogenic dose of 300 mg of nitrates. In this case, we would speculate that nitrate supplementation would not have increased performance as significantly compared to other participants as the baseline nitrate intake was already approaching the 300 mg nitrate intake. However, this was not the case, as participant 3 covered 2.7 km and 2.95 km in the Cooper test in the placebo and intervention groups, respectively, placing him on the higher end of the confidence interval spectrum. This alone would indicate that nitrate supplementation is useful if the baseline nitrate intake from whole foods is below 300 mg, regardless of what that baseline intake is, and that there is no dose-dependent effect of nitrate intake below the ergogenic dose. However, individual results from certain other participants do not support this notion. For example, participant 1 (Figure 2) had an average dietary nitrate intake of 71.5 mg for the placebo treatment and 79.4 mg for the intervention treatment, which was the lowest nitrate intake within the intervention trial. In this case, we would speculate that nitrate supplementation would have increased performance the most significantly; however, this was not the case, as participant 1 covered 3.5 km and 3.52 km in the Cooper test in the placebo and intervention groups, respectively. There are several reasons as to why these effects were observed. For instance, participant 1 had the longest covered distance in the Cooper trial in both the placebo and intervention groups. As such, participant 1 had the highest average estimated VO2max value of 67.2 mL/kg/min. Previous research indicates that nitrate supplementation is less effective in well-trained individuals (VO2max > 65 mL/kg/min) [2,19]. Perhaps this could explain why the addition of nitrates did not have a significant effect on performance in participant 1; however, it did have a significant effect on participant 3, who had an estimated VO2max value of 52 mL/kg/min. If this is the case, then this would also support the notion that the effects of nitrates are not as significant in well-trained individuals due to these athletes not being as reliant on nitric oxide production via the nitrate–nitrite–nitric oxide pathway by having higher baseline nitrate levels when compared to less trained individuals [4,20] and not by having higher baseline levels of nitrate intake through nutrition, as participant 1 had an approximately three-times-lower nitrate intake than participant 3. Another explanation is that a dose-dependent effect of nitrate intake below the 300 mg threshold exists, but only for less trained individuals and/or the effects are so low it might not be practically relevant. However, this is only speculation based on the available data of this study. Caution is advised when interpreting these findings due to the small sample size, and an intervention with a higher number of highly trained participants is needed in the future to give a further insight into these questions.

Comparing the performance within subjects with a high baseline intake of nitrates (>300 mg) was meant to give a further insight into the relationship between dietary nitrate intake and performance. We hypothesized that nitrate supplementation would not have a significant positive effect on covered distance in the Cooper performance test compared to the placebo if the baseline dietary nitrate intake was high (>300 mg nitrates) in the placebo group, as this would indicate that participants had already ingested enough nitrates through their diet to elicit ergogenic benefits. However, the sample size was too small to obtain a definitive answer. Looking into the results of the two participants (participant number 14 and 15, Figure 3) who were eligible for this comparison, it seems that our assumption was correct. The participants had an average covered distance in the Cooper test of 3.02 km and 3.01 km in the placebo and intervention trials, respectively. The average nitrate intake of these two participants in the placebo trial was 335 mg, just above the 300 mg ergogenic threshold. Both participants had an estimated VO2max value below 65 mL/kg/min, indicating that training status should not have affected the results. Previous studies have examined the effectiveness of nitrates on different aspects of physical performance [1,2,3,4]; however, to our knowledge, no studies have reported baseline nitrate intake prior to the start of the intervention. This is be an important variable that should be controlled for in future research. As such, prospective research with a higher number of participants is needed to give a further insight into the effectiveness of nitrate supplementation when the baseline nitrate intake is already at an ergogenic dose.

The average nitrate intake from whole foods from football players in this study was 165 mg. This is a greater intake than that reported by Jonvik et al. (2017), where the median dietary nitrate intake was 106 mg in a large group (n = 553) of highly trained Dutch athletes [14]. However, these were not only football players, but also athletes from other strength and sprint sports, team sports, and endurance sports. The highest reported nitrate intake without supplementation in our trial was 370 mg, meaning that no football player consumed more than 400 mg of nitrates, in comparison to 10 athletes in the Jonvik et al. (2017) trial, who had exceeded this amount [14]. These two athletes who consumed more than 300 mg of nitrates in this study had, on average, eaten two cups (approximately 150 g) of lettuce per day, with usually one cup at lunch and the other at dinner, accompanied by chopped vegetables such as paprika, carrots, and tomatoes. Interestingly, no other vegetables other than these were consumed by the athletes, apart from garlic and onions in certain cases, indicating quite a low heterogeneity in foods eaten by the football players. This vegetable intake is in line with results and guidelines provided in previous research [13,14]. To our knowledge, no other study has reported daily nitrate intake in athletes from whole foods. When compared to the population as a whole, football players in this study tended to be on the higher end of the spectrum of estimated daily nitrate intake from whole foods compared to the numbers reported in Europe (estimated range: 31–185 mg/d) and well over the numbers reported in the United States (estimated range: 40–100 mg/d) [15]. Taken together as a whole, this would indicate that the football players in this study had above-average nitrate intake comparatively speaking, with most of these nitrates coming from nitrate-rich vegetables and the rest from drinking water, fruits, and cured meats to a certain extent. It seems that the football players, on average, covered more than half of the required ergogenic nitrate dose without paying particular attention to their nitrate intake. As mentioned, most of the players’ nitrate intake came in the form of nitrate-rich leafy vegetables during their obligatory lunch at the training facility. The rest of the nitrate intake was due to drinking water, fruit, and cured meats in certain cases. Perhaps being more mindful of leafy or root vegetable intake in general during lunch and dinner and/or adding another vegetable serving to one of the main meals could be sufficient for most, if not all, of these football players to achieve an intake of nitrates close to or above 300 mg. This might be an effective strategy throughout most of the training period; however, it might not be the best option for matches. The half-life of nitrate is estimated at 5–8 h [9,32], meaning that attaining sufficient nitrate intake from diet alone might not be manageable if there is a match earlier in the day and since athletes might want to avoid higher fiber intake pre-match and opt for more easily digestible carbohydrates to reduce the occurrence of gastrointestinal issues [33]. A dietary nitrate supplement might be the better alternative in these pre-match situations. However, it is worth noting that certain studies have observed a return to baseline of saliva and plasma nitrate up to 24 h [34,35] after acute supplementation, with Kapil et al. (2015) reporting that salivary, plasma, and urinary nitrate levels returned to baseline levels a whole two weeks after a four-week supplementation with beetroot juice in otherwise hypertensive individuals [36]. This might indicate that acute nitrate supplementation pre-match might not be necessary, provided a high nitrate diet is kept regularly by the athlete. However, with the absence of higher-quality literature at this point, we again caution the interpretation of these results.

Perceived exertion was measured using a 6-to-20-point Borg scale [26] at four different time points (at each quarter of the trial) throughout both the intervention and placebo trials and then compared to one another. There were no statistically significant differences in perceived exhaustion between the placebo and intervention groups. This is an interesting finding, as participants in the intervention trial improved performance over the Cooper test when the baseline nitrate intake was below 300 mg. Altogether, this could indicate that the football players had improved performance when supplementing with nitrates; however, they did not subjectively feel more tired doing so. When compared to other research, these findings are in line with results reported in a systematic review and meta-analysis by Gao et al. (2021), where nitrate supplementation did not have a positive effect on perceived exertion but did have positive effects on endurance-based performance [37].

The average VO2max of the participants was 54.1 mL/kg/min, with the lowest recorded value being 44.1 mL/kg/min (playing position: goalkeeper) and the highest recorded value being 67.4 mL/kg/min (playing position: midfielder). This range is quite high; however, it was not unexpected due to the different playing positions of the football players. VO2max is often regarded as the best indicator of aerobic capacity [38]. The results of this study fall just below the line of average results reported in previous research of VO2max values between 55 and 65 mL/kg/min [38,39,40,41,42]. Interestingly, these results indicate that many of the high-level football players in these studies could be classified as “less trained” by the VO2max cutoff point of 65 mL/kg/min, where nitrate supplementation seems to be less effective as described in previous research [2,19]. This might be true strictly in the sense of aerobic capacity; however, football is a very complex sport where performance is not based on endurance alone, but also on speed, strength, agility, coordination, and through technical, tactical, and physical skills [41]. Only one football player in the current study had a VO2max value above 65 mL/kg/min and his performance remained unchanged when nitrates were supplemented despite the relatively low bassline nitrate intake, as discussed in the previous sections of this paper. The rest of the players’ performance should not have been affected by VO2max as the values were below 65 mL/kg/min.

Noteworthily, none of the athletes in this study had previously used beetroot juice or nitrate supplements of any kind prior to the study. Some of the athletes reported having heard that beetroot juice could be beneficial but had no knowledge of how and why beetroot juice should be or could be used in the context of improving sports performance. This highlights the need for more structured sports nutrition counseling in the First Division of Slovenian football, which is in line with our previous research on nutrition in elite football players [43].

### Study Limitations

There are several limitations to this study. Firstly, the results of this study refer to a small group of Slovenian football players aged 19–31. A strong limitation of this study was also the small sample size in regard to examining the effects of nitrate supplementation when the baseline nitrate intake already exceeds 300 mg (Hypothesis 2). The limitations also include the reliability and validity of the estimation of average dietary intake. As with most food tracking in research, some participants may have under-reported, and others may have over-reported, their intake of certain foods. However, the research team did their best to minimize these effects by providing visual cues for food logging and assembled dishes as well as working with each participant individually to report their nutrition as accurately as possible. Another possible limitation includes the calculation of nitrate intake from participants’ food diaries. We included data based on the PRODI^®^ 6.4 Expert program; however, we do realize that the nitrate content of certain foods might differ substantially using other software. A solution here might be using uniform analyzing software regarding nitrate content evaluation throughout this research area. There are also strides being made by other researchers to compile more comprehensive food databases of nitrate in nitrite reference values in both plant-based foods [44] as well as animal-based foods [45], which could be used in future research as a standard for dietary nitrate evaluation. Another limitation of the study was the estimation of the VO2max of the athletes via the Cooper (1968) equation rather than an actual measurement, which might have affected some of the drawn conclusions. When possible, direct VO2max measurements, rather than estimations based on equations, should be used. Related to this, sample homogeneity could also be treated as a limitation. While we expected quite a homogenous population as the football players were from the same football club and going through the same training regimen, there were apparent differences in their fitness levels depending mainly upon their playing position, which might have influenced the results. A resolution to this problem is the same as for the sample size problem for Hypothesis 2 testing: a larger pool of football players, possibly from different football clubs, but competing at the same level (e.g., Slovenia’s First Division) and ideally playing in the same position, should be included in future research. Lastly, body composition parameters were estimated using BIA-based predictive equations with oscillating values depending on the formula applied [46]. Generalized BIA-based predictive equations overestimated FM% in athletes, and, therefore, the use of sport-specific predictive equations is strongly suggested [47]. Future research should aim for more reliable body-composition-measuring equipment, such as a dual-energy X-ray absorptiometry (DEXA) device [48], which is regarded as the gold-standard technique for body composition measurement.

## 5. Conclusions

In conclusion, this study showed that nitrate supplementation had a statistically significant positive effect on performance if dietary nitrate intake was below 300 mg. No definitive dose-dependent effect of nitrate intakes below the 300 mg ergogenic dose could be drawn from this study. No effects on the influence of nitrate supplementation on performance when the baseline nitrate intake was higher than 300 mg could be observed as the sample size was too small. There was no statistically significant effect of nitrate supplementation on perceived exertion during exercise, despite nitrate supplementation having a statistically significant effect on performance. The average nitrate intake of Slovenian First Division football players was 165 mg, coming mostly from leafy vegetables. Increasing vegetable intake seems to be a feasible way of attaining the ergogenic nitrate dose without having to use nutritional supplements.

## Figures and Tables

**Figure 1 nutrients-15-03721-f001:**
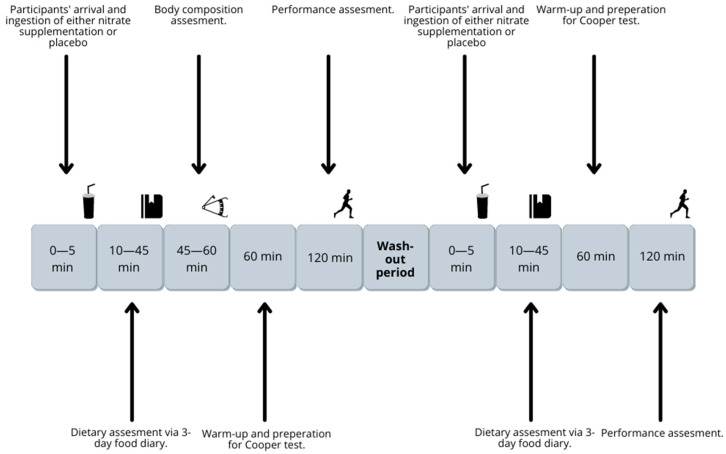
Study overview.

**Figure 2 nutrients-15-03721-f002:**
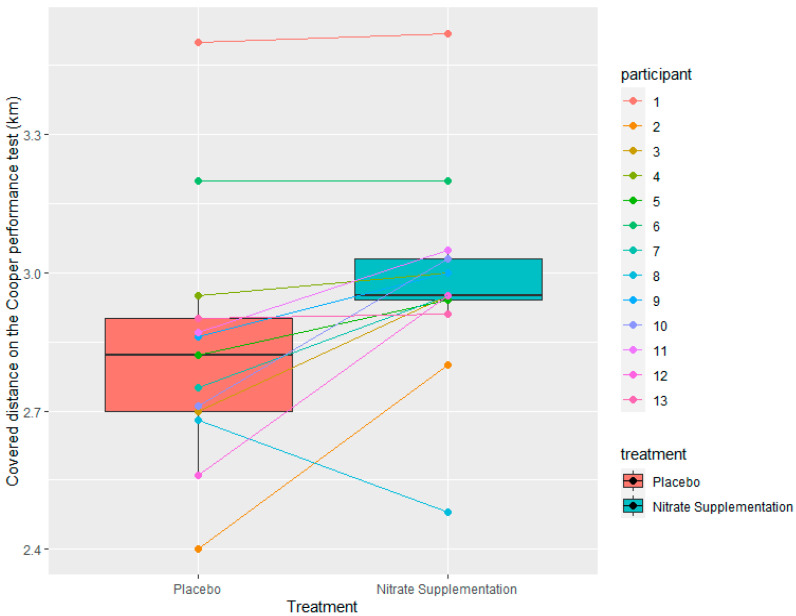
Distribution of distance covered in the Cooper test by placebo and intervention (*n* = 13). Intervention group: supplementation with 400 mg of nitrates. Placebo group: without nitrate supplementation.

**Figure 3 nutrients-15-03721-f003:**
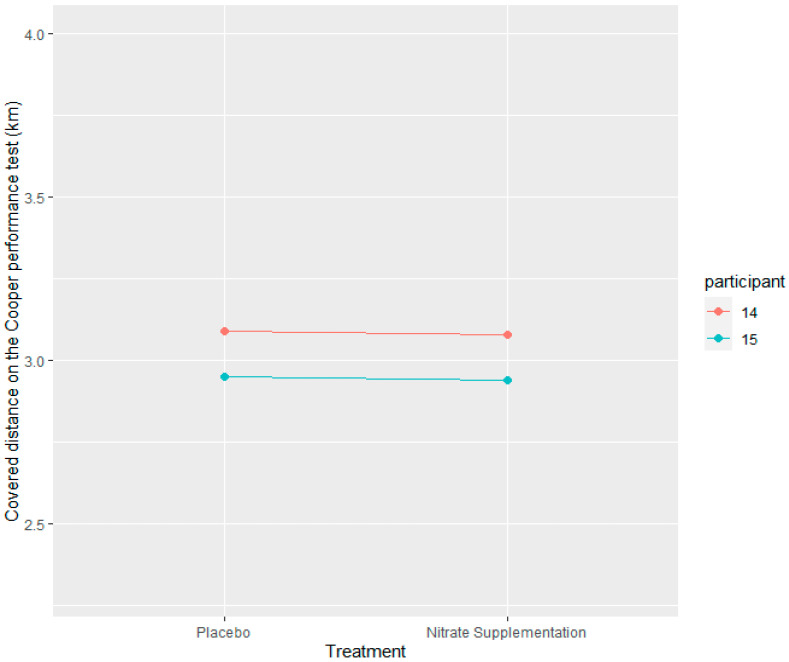
Distribution of distance covered in the Cooper test by placebo and intervention (*n* = 2). Intervention group: supplementation with 400 mg of nitrates. Placebo group: without nitrate supplementation.

**Figure 4 nutrients-15-03721-f004:**
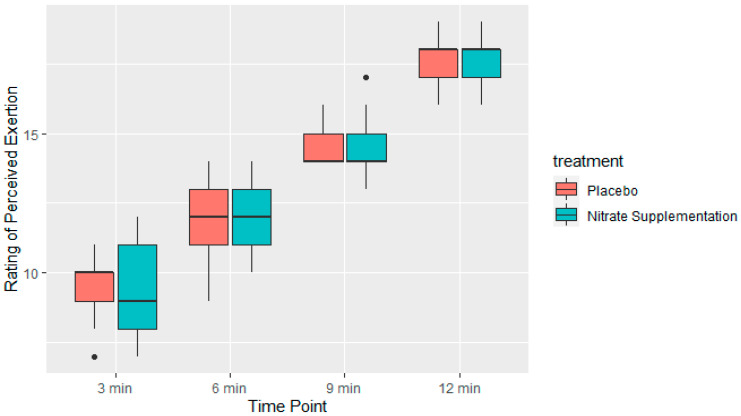
Average rating of perceived exertion during each quarter of the Cooper test (*n* = 15). Rating of perceived exertion was based on the 6-to-20-point Borg scale [26]. Intervention group: supplementation with 400 mg of nitrates. Placebo group: without nitrate supplementation.

**Table 1 nutrients-15-03721-t001:** Participant characteristics.

Variable	Mean ± SD	Range
Age (yrs.)	23.3 ± 2.7	19–31
Height (cm)	182 ± 7.4	169–192
Body mass (kg)	77.9 ± 8.6	60.5–88
BMI (kg/m^2^)	23.3 ± 1.2	20.5–24.9
FFM (kg)	60.2 ± 5.0	49.4–66.7
FM (kg)	17.6 ± 4.3	8.7–24.2
FM (%)	22.4 ± 3.6	13.9–27.5
VO2max (mL/kg/min)	54.1 ± 5.5	44.1–67.4

SD—standard deviation; BMI—body mass index; FFM—fat-free mass; FM—fat mass; VO2max was estimated based on Cooper performance test and Cooper equation (Cooper, 1968) and reported as an average of intervention and placebo trial.

**Table 2 nutrients-15-03721-t002:** Nutritional intake of subjects.

Variable	Mean ± SD	Range
Energy intake, PLA (kcal)	2732 ± 648	2030–4484
Energy intake, NO (kcal)	2733 ± 581	1966–4158
CHO intake, PLA (g/kg BM)	3.7 ± 1.6	2.0–8.4
CHO intake, NO (g/kg BM)	3.7 ± 1.7	2.3–9.6
Protein intake, PLA (g/kg BM)	1.8 ± 0.5	1–2.6
Protein intake, NO (g/kg BM)	1.8 ± 0.6	0.9–2.5
Fat intake, PLA (% daily energy intake)	33.1 ± 5.8	25–47
Fat intake, NO (% daily energy intake)	34.8 ± 4.6	29–45
Average nitrate intake (mg)	165 ± 75	71.5–370.3
Nitrate intake, PLA (mg)	169 ± 83	71.5–370.3
Nitrate intake, NO (mg)	161 ± 70	79.4–275

SD—standard deviation; BM—body mass; PLA—placebo group; NO—intervention group; CHO—carbohydrate.

## Data Availability

Not applicable.

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
