# Peer review of "The Effects of Nitrate Supplementation on Performance as a Function of Habitual Dietary Intake of Nitrates: A Randomized Controlled Trial of Elite Football Players"

_nutrients, 2023, doi:10.3390/nu15173721_

Round 1
Reviewer 1 Report
TITLE AND ABSTRACT
- The title of the manuscript is concise, specific and relevant.
- The title clearly identifies that the study reports data from human trials, specifically from elite soccer players. In addition, it also concisely identifies the type of study that the authors have used to perform this work, in this case we are before a randomized trial as clearly read in the title.
- The title includes the study population, elite soccer players. However, it would be interesting to know the exact age range of these players, as well as their nationality, as shown later in the abstract where it does indicate that the players used for this study are of Slovenian nationality. Therefore, a suggestion to improve this current title could be: The effects of nitrate supplementation on performance as a function of habitual dietary intake of nitrates: a randomized controlled trial in Slovenian elite soccer players aged 19-31 years.
- The abstract is well structured as it is written in a single paragraph and has no headings, but the subsections of the trial design, methods, results, and conclusions are indicated in the abstract. It highlights the purpose of the study, the main test performed (Cooper's test) by the players, as well as the number of the sample used (n=15), the results obtained and their respective conclusions.
- In addition, the words that compose this section are adjusted to those indicated by the journal where it is published, not reaching the limit of 200 words.
- In the abstract the results shown are present and substantiated in the main text and the main conclusions are not exaggerated.
KEYWORDS
- The keywords are correct; they have to be three to ten relevant keywords after the abstract. In this study there are 3 keywords specific to the article and common within the discipline of the subject such as: nitrates, dietary supplements and sport performance. However, it would be advisable to add some more.
INTRODUCTION
- The introduction places the study in a broad context and highlights its importance because nitrates increase endurance performance by decreasing oxygen cost during submaximal aerobic exercise and increase power production through improved muscle contractile function.
- Line number 100 of the article defines the purpose of the work and its importance; in addition, it also states the specific hypotheses being tested as we can see from line 104 through 113 with those three hypotheses being stated by dots.
- The introduction is kept understandable for scientists working outside the subject of the article.
- Most of the references used are recent and follow the subject of the study with the exception of reference number 8 and 21 which are much older.
- In general, the introduction is quite complete and shows a good overview of the current state of the subject. However, as observations to be highlighted, it would be necessary to indicate at the end of the introduction the main objective of the work and highlight the main conclusions.
MATERIAL AND METHODS
- The type of study indicated is: randomized placebo controlled double-blind crossover trial. However, the mechanism used to implement the randomization sequence is not indicated, describing the steps taken to conceal the sequence until the interventions were assigned.
- The criteria for selection and exclusion of participants are indicated, as well as the sample size; however, it is not specified how this size was determined and why it was decided to opt for this number of individuals, which is apparently too few to draw accurate conclusions. In randomized clinical trials with a small sample size, it is possible to reach the erroneous conclusion that there are no differences between the comparison groups.
- It does not indicate who generated the randomization sequence, who selected the participants, and who assigned the participants to the interventions.
- The materials used during the study are well detailed throughout the section allowing others to replicate it exactly the same, such as the manufacturer of the beet juice concentrate consumed by the players both the one containing nitrates as well as the manufacturer of the placebo that looked and tasted similar to the supplement mentioned above, but without nitrates, the brand and model of the bioimpedance scale used to measure body composition and the model of the treadmill used prior to the Cooper test.
- Detail the statistical analysis, this section is very brief and does not mention the statistical methods used to compare the groups in terms of the primary and secondary response variable.
RESULTS
- A table with the baseline demographic and clinical characteristics of the participants is shown, however, a flow chart is strongly recommended for each group, those who took the beet juice concentrate and those in the placebo group, indicating the number of participants who were randomized, who received the proposed treatment and who were included in the main analysis.
- Indicate in the same flow chart mentioned above for each group, the losses and exclusions after randomization for each hypothesis, together with the reasons.
- Do not indicate the dates defining the recruitment periods of the players or the dates when the appropriate tests were performed for the study.
- The results for each group, the estimated effect size and its precision as 95% confidence intervals should be indicated. It is preferable to express the results as 95% CIs rather than as p-values, as these intervals give a truer idea of the magnitude of the observed differences and their clinical significance.
DISCUSSION
- Further comparison with relevant studies regarding the effects nitrates have on performance in elite soccer players would be needed (whenever possible include systematic reviews) Only the study by Jonvik et al. (2017), Gao et al. (2021), Kapil et al. (2015) is shown.
- It presents a number of limitations the study, since, when looking at the individual numbers, some contradictory evidence is observed, this is due to the small sample size.
- The various limitations presented by the study are indicated.
- Some method used to minimize some of these limitations is indicated, for example, to prevent players from underestimating or overestimating the intake of certain foods, visual cues were provided for recording food and assembled plates, in addition to working with each participant individually to report their nutrition as accurately as possible. However, it would be advisable to indicate possible solutions also for the other limitations indicated in this section.
CONCLUSION
- Conclusion (optional): has been included. The researchers have made a brief conclusion in this section. The decision to opt for a conclusion is totally correct due to the length and complexity of the discussion section.
REFERENCES
- In the references of journals, the volume of the articles is observed in bold when it should not be so, since in bold has to be the year of publication of that study. Example of the guide of the journal of the study, to introduce the references correctly in the references section of the work:
o Journal Articles: Author 1, A.B.; Author 2, C.D. Title of the article. Abbreviated Journal Name Year, Volume, page range.
- Another aspect to be taken into account in this section of the work concerning the reference, specifically those that belong to journal articles, and as mentioned in the previous point in the example, is that the journal in the reference should appear in abbreviated form. However, in the present work it is observed that in some references the journal is indicated with its respective abbreviation, while in other references it is not in its abbreviated form. They should all be abbreviated. In some references it is observed how the journals: Nutrients (reference no. 1), The Journal of biological chemistry (reference no. 8) or International journal of sport nutrition and exercise metabolism (reference no. 11) are not abbreviated. However, others such as: Medicine and Science in Sports and Exercise (Med. Sci. Sports Exerc.) The Journal of Strength and Conditioning Research (J. Strength Cond. Res.) or Nature Reviews Drug Discovery (Nat. Rev.Drug Discov.). They must all be abbreviated.
Author Response
Dear reviewer,
Thank you for the time taken for this extensive review, it is greatly appreciated! We have taken into consideration everything outlined in your review and have made the appropriate accommodations:
1) Research title has been changed.
2) Keywords have been added.
3) At the end of the introduction we have added the main objective of the work and highlighted the main conclusions.
4) Materials and methods
- In the beginning of the methods section we have added details surrounding blinding and randomization.
- Explanation regarding sample size calculation has been added in the 2.1 Subjects section. To further clarify, according to our calculation, the sample size should have been enough to elicit a meaningful finding in all hypotheses based on the sample size calculation. However, we had wrongfully speculated that the football players would generally have a “healthier” diet with a higher vegetable intake which included more nitrates and that we would have more eligible participants for Hypothesis 2. This was unfortunately not the case. Even though the participants of the study had on average eaten nitrates on the higher end of the spectrum compared to the general population as highlighted in the Discussion section, they were still far below the set 300 mg nitrates mark, and we were left with only two participants for H2.
5) Results
- In the results section, we added the recruitment period, intervention dates, and confidence intervals.
- As all 15 participants underwent randomization and successfully completed the experimental trial (i. e. they had completed the Cooper trial with a nitrate supplement as well as with the placebo), meaning there were no exclusions, a flow chart was deemed unnecessary.
6) Discussion
- Further comparison with relevant studies regarding nitrates' effects on performance specifically in elite football players was added, although research is sparse in this regard.
- Possible solutions for other limitations indicated in the discussion section have been added.
7) References have been corrected.
Every change can be seen in the revised document. We hope to have addressed all of the comments to the reviewers' satisfaction.
Kind regards,
Matjaž Macuh
Reviewer 2 Report
I read the manuscript entitled "The Effects of Nitrate Supplementation on Performance Depending on Habitual Dietary Nitrate Intake: A Randomized Controlled Trial in Elite Football Players" with great interest. Personally, I found them well written and organised. There are just some points to clarify:
- Line 123: It would be useful to provide more information on the randomization and blinding here. For instance, how was blinding performed, or who is blinded? (See doi.org/10.3390/medicina57070647 and doi.org/10.1186/s13063-020-04607-5)
-Line 134: The sample size determination needs to be clarified; how did you establish that?
-line 199: Regarding fat free mass, muscle mass, and fat mass, you should highlight that the gold standard (such as magnetic resonance for quantifying muscle mass or DXA for fat mass) has not been used to quantify those parameters, and results depend on BIA-based predictive equations with oscillating values depending on the formula applied (doi:10.3390/nu15051160). Which formula have you used to obtain FFM and FM? The choice of the equation determines the validity of the FM% prediction. This is an important consideration (see here: doi.org/10.3390/nu15020278) that you have to mention.
Since you have raw parameters (reactance, resistance, and phase angle), put them into a graph using the BIVAs approach, showing the main vector of your population. Remember that BIVA does not depend on equations (doi:10.1123/ijspp.2013-0119).
However, using the BIVA approach, there is a link between body composition and aerobic power (see here: doi.org/10.3390/biology11040505), and you should investigate the relation between phase angle and VO2max in your sample.
Author Response
Dear reviewer,
Thank you for the compliment and the time taken to give us constructive and useful feedback, it is greatly appreciated! We have taken into consideration everything outlined in your review and have made the appropriate accommodations:
- In the beginning of the methods section we have added details surrounding blinding and randomization.
- Explanation regarding sample size calculation has been added in the 2.1 Subjects section.
- In the methods section (previous line 199) we have added details surrounding BIVA measurements and the prediction equations used, which is the Bodygram PLUS Software V. 1.0 (Akern Srl., Pontassieve, Florence, Italy) which has been used in previous research (https://www.mdpi.com/1660-4601/17/3/729). We have reached out to the company but have not been given more specific details surrounding the equation in the software. We have also added a section at the end of the Discussion alongside other study limitations explaining the limitations of using BIVA as a method of body composition assessment and highlighting DXA as the gold standard.
- Regarding the graph of the main vector of our population using the BIVA approach we are currently experiencing some issues. As you may have been informed by the Nutrients Editorial Office we had experienced serious flood issues in Slovenia in the past week, and the bodygram software on my laptop was lost during this time. However, I do have backed up data as well as raw data in Excell for all participants and can easily produce the graph as soon as our technician gets back from their holiday, and I am granted access to another bodygram software. I am afraid, however, that this will not be possible until the end of August. If there is a possibility that we add the graph post-hoc or this manuscript stays pending until then, we are happy to make the adjustments necessary.
- Regarding the last point of investigating the link between body composition and aerobic power, and the relation between phase angle and VO2max:
These are all great comments; however, we believe these analyses are beyond the scope of this paper and that they would differ away from the main objective of this study, which was to investigate the effects of nitrate supplementation on performance depending on habitual dietary nitrate intake. The purposed analyses could be a separate research paper on its own as there are many other variables that could be delved into on top of the ones mentioned. For this specific research paper, we just wanted to include basic information regarding the body composition of the participants.
We have, however, explored some of these relationships regarding body composition and performance in our previous work on a similar sample of athletes: https://doi.org/10.3390/nu15010082
We hope to have addressed all of the comments to the reviewers' satisfaction.
Kind regards,
Matjaž Macuh
Round 2
Reviewer 2 Report
Well done, then I have just some suggestions:
-Regarding the title, personally, I don't think it is important to specify the nationality and age of participants; maybe it sounds better: "The effects of nitrate supplementation on performance as a function of habitual dietary intake of nitrates: a randomized controlled trial in elite football players". In the limitation, you could write that your results refer to a small group of slovenian football players aged 19-31... In addition, I strongly recommend separating limitations from discussion.
-if you have raw data (Rz, Xc and PhA), you can upload a file as supplementary materials, then, lines 238-244 should be modified as follow:
“Body mass and height of the subjects were measured using a scale with a stadiometer. Based on these data, the body mass index (BMI) was calculated. Participants underwent bioelectrical impedance vector analysis (BIA 101 BIVA® PRO, Akern, Pisa, Italy) to obtain body composition parameters. Fat mass (FM), fat-free mass (FFM), and total body water (TBW) were obtained using Bodygram PLUS Software V. 1.0 (Akern Srl., Pontassieve, Florence, Italy). Furthermore, as BIVA represents a qualitative analysis (doi.org/10.1016/j.clnu.2023.07.025), the raw data, such as resistance and reactance of each participant, are available as supplementary material” or something similar…
-Lines 544-549 should be modified as follow:
“Lastly, body composition parameters were estimated using BIA-based predictive equations with oscillating values depending on the formula applied [49]. Generalized BIA-based predictive equations overestimated FM% in athletes, and, therefore, the use of sport-specific predictive equations is strongly suggested (cite doi.org/10.3390/nu15020278). Future research should aim for more reliable body composition measuring equipment, such as a dual-energy X-ray absorptiometry (DEXA) device [50], which is regarded as the gold standard technique for body composition measurements”
Finally, of course, check the acronyms and bibliography.
Author Response
Dear reviewer,
Thank you for your further insights and helpful suggestions!
We have integrated all the provided comments into the revised manuscript and will upload the raw anthropometric data as supplementary material.
We hope to have addressed all of the comments to the reviewers' satisfaction.
Kind regards,
Matjaž Macuh